# *Lilium brownii*/*Baihe* as Nutraceuticals: Insights into Its Composition and Therapeutic Properties

**DOI:** 10.3390/ph17091242

**Published:** 2024-09-20

**Authors:** Yong-Fei Wang, Zi-Yi An, Le-Qi Yuan, Ting Wang, Wei-Lin Jin

**Affiliations:** 1The First Clinical Medical College, Lanzhou University, Lanzhou 730000, China; wyongfei2024@lzu.edu.cn (Y.-F.W.); 120220903500@lzu.edu.cn (Z.-Y.A.); yuanlqzbc@126.com (L.-Q.Y.); tingwang2023@lzu.edu.cn (T.W.); 2Institute of Cancer Neuroscience, Medical Frontier Innovation Research Center, The First Hospital of Lanzhou University, The First Clinical Medical College, Lanzhou University, Lanzhou 730000, China

**Keywords:** anti-depression, anti-tumor, Baihe, nutraceuticals, Lilium, pharmacological effects

## Abstract

Nutraceuticals are compounds or components in food that offer health benefits. They can be incorporated into food to make it functional or used as supplements or medicine. *Lilium brownii*/*Baihe* is one of the classic nutraceuticals. The chemical composition of Lilium is complex and has a variety of pharmacological effects. Moreover, the compound preparation based on Lilium has been used in the treatment of respiratory diseases in traditional Chinese medicine. In addition, Lanzhou lily has become food on the dinner table. Therefore, *Lilium brownii*/*Baihe* is a nutraceutical with a long history. Based on the current understanding of Lilium, this review provides an in-depth discussion of the bioactive components and pharmacological effects of Lilium. This is important to provide theoretical reference for the in-depth study of Lilium as well as its development and application in medicine, food, and other industries.

## 1. Introduction

With the increasing improvement in people’s living standards, dietotherapy, medicinal diet, and health preservation based on the theory of nutraceuticals have developed rapidly and become the focus of attention [1]. Nutraceuticals or natural products are commonly called medical foods. The continuous intake of nutraceuticals has a beneficial effect on human health and can improve symptoms of certain diseases, such as cardiovascular diseases, diabetes, neurological diseases, and cancers [2]. The theory of nutraceuticals can be traced back to ancient times. In the process of looking for food, people gradually found that some natural animals and plants used for relief and treatment of diseases could also be used as food, which led to the theory of nutraceuticals. In China, it is included in the work of Sun Simiao’s “Qian Jin Yao Fang. Food treatment” [3], Meng Kui’s “Shi Liao Ben Cao” [4], Pu Zengguan’s “Bao Sheng Yao Lu” [5], and Hu sihui’s “Yin Shan Zheng Yao” [6]. The theory and application of nutraceuticals have been accumulated and enriched continuously.

*Lilium/Baihe* is one of the classic nutraceuticals, and its edible and medicinal parts are the bulbs of Lilium. Lilium has active components determined by pharmacological studies, which mainly include saponins, polysaccharides, flavonoids, and alkaloids. Lilium medicinally has the effect of nourishing yin and moistening the lungs, dispelling fire, and calming the mind, and it is often used to treat symptoms such as yin deficiency and chronic cough, fatigue, cough and hemoptysis, deficiency and palpitation, insomnia, dreaminess, and trance. Lilium can also be used in diet therapy for cough and asthma, hypoglycemia, anti-tumor, improved sleep, enhanced immunity, Alzheimer’s prevention, and other effects.

Therefore, *Lilium/Baihe* is a nutraceutical with a long history. Based on the current understanding of Lilium, this review provides an in-depth discussion of the chemical constituents and pharmacological effects of Lilium. This is important to provide theoretical reference for the in-depth study of Lilium as well as its development and application in medicine, food, and other industries.

## 2. *Lilium/Baihe*: The Brief Introduction

### 2.1. Lilium: Distribution and Habit

*Lilium brownii* is a plant in the Lilium family. The Lilium bulb is globose and scales lanceolate. The leaves are scattered and usually decrease from bottom to top, oblanceolate to obovate. The flowers are solitary or several and arranged subumbellate. The anthers are long and elliptic, the ovary is terete, and the style stigma is three-lobed. The flowering period is from May to June, and the fruiting period is from September to October. Most Lilium likes sunny environments, and a few varieties like shade; it can grow in semi-wetland, cold-resistant, cool, and humid climates, which are sandy loam or humus with deep, loose, fertile, moist, and well-drained soil [7]. Lilium is mainly distributed in the temperate and cold regions of the Northern Hemisphere, such as eastern Asia, Europe, and North America, but rarely in the tropics. It is reported that the annual output of Japanese and Korean Lilium is second only to China, which is the country with the widest distribution of wild Lilium resources and the most varieties [8]. There are about 100~175 species of Lilium wild resources in the world, mainly distributed in Asia (about 70 species), North America (about 24 species), and Europe (including Turkey and the Caucasus) in the Northern Hemisphere [9] (Figure 1). Lilium can be divided into seven groups according to bulb characteristics, leaf inflorescence, flower pattern, and origin: Martagon, Archelirion, Sinomartagon, Daurolirion, Leucolirion, Pseudolirium, and Liriotypus [10] (Table 1).

Lilium has a long history, and it is a complex process from planting to picking, processing, and finally being used as medicine and food, which condenses the wisdom of our ancestors and the efforts of modern scholars.

### 2.2. Lilium: Complex and Diverse Bioactive Components

The bioactive components of Lilium include steroidal saponins, polysaccharides, phenols, flavonoids, and alkaloids (Table 2). Steroidal saponins, polysaccharides, and phenols are the main bioactive components.

#### 2.2.1. Steroidal Saponins

Steroidal saponins are a class of natural products with diverse structures characterized by non-polar saponins linked to one or more polar carbohydrate structures. According to the structural differences of aglycone, steroidal saponins can be divided into four categories: spirosterol, isospirosterol, spirosterol, and furansterol. Steroidal saponins are rich in bulbs of Lilium species, accounting for about 13% of the total freeze-dried weight of Lilium bulbs [11]. At present, a total of 82 steroids have been extracted from Lilium bulbs [12]. The content of saponins is different in different varieties of Lilium [13].

#### 2.2.2. Polysaccharides

The structural description of polysaccharides includes the range of relative molecular weight, composition of monosaccharides, type of connection points, configuration of monosaccharides, and glycosidic bonds and repetitive units. At present, 13 kinds of Lilium polysaccharides purified have been preliminarily analyzed [14]. The content of polysaccharides is different in different varieties of Lilium [15].

#### 2.2.3. Phenols

Lilium is also rich in phenolic compounds. Phenols are a large class of bioactive components in the genus Lilium, such as Lilium glycosides, followed by phenolic compounds. So far, 46 phenolic compounds have been isolated from Lilium [14,16]. Flavonoids are the most widely distributed phenols. At present, 20 flavonoids have been isolated from Lilium, and the content of flavonoids is different in different varieties of Lilium [17].

#### 2.2.4. Alkaloid

Alkaloid is one of the bioactive components in Lilium. So far, eighteen kinds of alkaloids have been isolated, including three steroidal alkaloids [14]. The results showed that the concentration of steroidal glycoside alkaloids was higher than that of furostanol saponins in all structures except the roots of Lilium. The ratio of steroidal glycoside alkaloids and furostanol saponins was higher under light and decreased sequentially from aboveground organs to underground organs [18].

#### 2.2.5. Emodin

Lilium contains a small amount of emodin, which is an important component of Lilium activity [19].

#### 2.2.6. β Sitosterol

Lilium flower contains a small amount of β sitosterol, which is also an important component of Lilium activity [20].

#### 2.2.7. Other Components

In addition to steroidal saponins, phenolic glycosides, and polysaccharides, other glycosides and alkanes are extracted from Lilium bulbs. For example, Yan-hu Wen extracted adenosine from Solanum lanceolata [21]. Yoshihiro Mimaki extracted methyl α-D-mannopyranoside from L. tenuifolium [22]. In addition, the bulb of medicinal Lilium is also rich in amino acids, phospholipids, and dietary fiber.

The bioactive composition of Lilium is relatively complex. At present, people’s understanding of Lilium composition and the method of extracting Lilium bioactive components are still limited. The active development of new methods for extracting bioactive components is the basis for research on the edible and medicinal value of Lilium.

Furthermore, there are many varieties of Lilium, and Lilium grown in different geographic regions have different bioactive components, which may affect their medicinal properties. The main bioactive components of Lilium are saponins and polysaccharides. The planting range of Lilium is wide, and the quality of Lilium varies greatly among different areas. The growth of Lilium is significantly affected by specific ecological and climatic conditions, which are directly related to the quality of the Lilium [23]. A study found that temperature and rainfall are the key climate factors for the quality formation of Lilium, and high-temperature climates promote the accumulation of total polysaccharides in Lilium [24]. This geographical difference is closely related to the efficacy of the Lilium. Firstly, in terms of taste, Lilium lancifolium Thunb in Longshan Hunan, also known as “Longshan lily”, has a bitter flavor. The high saponin content of Longshan lily makes it a classic medicinal Lilium [25]. However, Lanzhou lily from Gansu Province is sweet and tasty and has a high polysaccharide content, making it a local specialty food [26]. In addition, there are significant differences in the bioactive components of a lily species in different geographical regions. A study analyzed the polysaccharide content in Lanzhou lily from six major origins in and around Lanzhou and found that the polysaccharide content of Lanzhou lily from different origins varied considerably [27]. Secondly, the medicinal properties of Lilium vary from one geographical location to another. Most Lilium is usually non-toxic and has therapeutic properties for a wide range of ailments. However, lily of the valley, which grows throughout Europe, North America, and Asia, can cause acute poisoning when consumed [28]. In addition, Lilium is mainly used to treat mastitis, liver disease, and herpes zoster in Europe [29,30,31,32]. In Asia, Lilium is mainly used to treat lung diseases [33,34,35,36], whereas in North America, Lilium is mainly used for food [37,38]. This may be related to the bioactive components contained in Lilium from different regions. No studies have yet reported on the specific differences in efficacy of Lilium across different geographical locations. Future research in this area will provide scientific basis for rational planning and cultivation of Lilium medicinal materials as well as sustainable utilization of resources.

**Table 1 pharmaceuticals-17-01242-t001:** The world distribution of Lilium.

Distribution	Species and Quantity	Representative Category	Growth Characteristics	Main Use	References
Asia	70	Martagon, Leucolirion, Archelirion, Daurolirion, Martagon	Well-drained slopes, mountain meadows, forest margins, rock crevices, forest slopes, thickets, grasses, and valleys	Antitussive and expectorant, diuretic, antipyretic.	[33,34,35,36]
North America	24	Pseudolirium	Streams and swamps	food	[37,38]
Europe	22	Martagon, Liriotypus	A place with high mineral coverage	Wound healing; treatment of mastitis, liver disease, and herpes zoster.	[29,30,31,32]

**Table 2 pharmaceuticals-17-01242-t002:** The main bioactive components of Lilium.

Bioactive Constituents	Quantity	Main Existing Site	Reference
Steroidal saponins	82	Flower buds, bulb	[13]
Polysaccharides	13	Bulb	[15]
Phenols	46	Bulb	[17]
Alkaloid	18	Bulb	[18]

### 2.3. Lilium: Both Food and Medicine

At present, a balanced diet and healthy food have received more and more attention because of their great impact on health; Harvard Law School recommends integrating food and nutrition into health care [39]. It is worth pointing out that nutraceuticals play an increasingly important role in maintaining human health. Lilium is a kind of nutraceutical, which contains a variety of functional compounds with pharmacological activities.

The traditional uses of Lilium are diet therapy and medicine. Many diet therapies based on Lilium bulbs have been recorded in ancient Chinese books for thousands of years. There is a quote in Ben Cao Gang Mu: “The new Lilium can be steamed and boiled, and meat is better, and the dry is best for powder food”. Lilium tastes sweet and slightly bitter, so one can make a Lilium soup or Lilium porridge and add a little sugar to neutralize its bitter taste in the summer, which plays a role in dispelling fire, relieving summer heat, and moistening the lungs. There are also Lilium-fried celery and Lilium-fried meat slices, which are all famous Chinese dishes.

Lilium has a long history, and its medicinal value has been well documented since ancient times. The most classic is that it has the effect of nourishing yin and moistening the lungs, dispelling fire, and calming the mind. In addition, Lilium also has a therapeutic effect on the main yin deficiency cough for a long time, blood in sputum, late-stage fever, residual heat that is not clear, or emotional issues caused by virtual annoyance, palpitations, insomnia, dreaming, trance, carbuncle swelling, and eczema. In ancient times, many compatibilities and attached prescriptions of Lilium played an important role in Chinese traditional medicine. The earliest record of the use of Lilium as medicine is Shen Nong Ben Cao Jing: “Lilium, which treats evil qi, abdominal distension, heartache, facilitates defecation, and replenishes qi.” Since then, many ancient books have medicinal records of Lilium, such as Baihe Gujin decoction (BGD): “Nourishing Yin and moistening Lung, resolving phlegm and relieving cough, treating Lung and Kidney Yin deficiency”; Baihe Zhimu decoction (BZD): “Calming the mind, moistening the lung and dispelling fire, treating Lilium disease after mistakenly sweating, body fluid injury, deficiency heat aggravation, and heart thirst”; Baihe Jizihuang Tang (BHT): “Nourishing yin and eliminating annoyance, treating Lilium disease, empty and restless, dry lips, and dry stool”; and Baihe Dihuang decoction (BDD): “Moistening the heart and lungs, dispelling fire and cooling blood, treating Lilium disease”. Continuing from traditional medicine, the 2015 edition of Chinese Pharmacopoeia includes proprietary Chinese medicines such as Baihe Gujin Oral liquid, Baihe Gujin Pill, and Honey Lilium, which play a certain role in the treatment of lung disease.

The traditional use of Lilium is the crystallization of the wisdom of the ancient working people and the valuable experience of Lilium as a nutraceutical, which lays a foundation for the study of the pharmacological effects of Lilium from the perspective of modern medicine.

## 3. Lilium: Pharmacological Effects and Mechanisms of Action

Lilium has a wide range of pharmacological effects (Figure 2), which is related to its composition. Lilium may inhibit inflammation, remove oxidation products, and improve the tolerance of the body to oxidation products through bioactive components such as polysaccharides, amino acids, alkaloids, phenols, and flavonoids, which modulate the immune, endocrine, nervous, and metabolic systems. On the other hand, the effects of anti-tumor, anti-bacterial, and regulation of the brain–gut axis are supplements of modern pharmacological research to the new efficacy of Lilium [40]. The compound preparations based on Lilium also have a wide range of pharmacological effects. In the following sections, the plant pharmacological effects of Lilium are introduced from two aspects: Lilium bioactive components and compound preparation based on Lilium.

### 3.1. Anti-Tumor

With an in-depth understanding of anti-tumor drugs, the discovery of the anti-tumor effect of traditional Chinese medicine provides people with new ideas and strategies for the treatment of tumors. In the treatment of cancer, plant secondary metabolites are well classified as chemical defenses and are considered to be bioactive compounds for primary and secondary prevention [41,42,43]. Secondary metabolites can be divided into alkaloids, terpenoids, flavonoids, lignans, steroids, curcumin, saponins, phenols, and glycosides [44]. These compounds regulate metabolism and signal pathways, thereby controlling angiogenesis and inhibiting the formation of intracellular microtubule assembly and apoptosis [45]. In traditional Chinese medicine, Lilium, which is a nutraceutical, has an anti-tumor effect in many kinds of tumors. Some clinical or preclinical studies have shown that Lilium prescription or food therapy prescription has a good effect on the adjuvant treatment of cancer, which can increase the sensitivity of chemotherapy and improve the effect of chemotherapy and the quality of life of patients. At the same time, it can reduce the clinical symptoms and side effects of chemotherapy. The anti-tumor components of Lilium may be polysaccharides, saponins, and alkaloids, and its anti-tumor mechanism may be related to regulating oncogenes and hindering cell proliferation. In terms of tumor prevention and treatment, Lilium is also used to treat malignant tumors such as lung cancer, nasopharyngeal carcinoma, skin cancer, malignant lymphoma, and postoperative patients with lung cancer, especially the symptoms such as weakness and fatigue, dry mouth upset, less dry cough and sputum, hemoptysis, palpitation, and insomnia after radiotherapy. It is worth mentioning that a large number of cell and animal experiments on the anti-tumor effect of Lilium have been reported, but there are few clinical studies.

#### 3.1.1. Anti-Tumor Effect of Polysaccharide

Some studies have confirmed that Lilium polysaccharides inhibit the growth of subcutaneously transplanted H22 hepatocellular carcinoma by down-regulating the expression of bcl-2 protein, up-regulating the expression of Bax protein, and activating caspase-3 and caspase-9. This indicates that Lilium polysaccharide plays a role in inducing cell apoptosis by regulating oncogenes and mitochondria, which may be one of its anti-tumor mechanisms. Also, in hepatocellular carcinoma, some studies have confirmed that Lilium polysaccharide can promote apoptosis and exert its anti-tumor effect by down-regulating the expression of cyclinD1 [40]. In addition to hepatocellular carcinoma, it has been confirmed that Lilium polysaccharide can significantly inhibit the growth of subcutaneously implanted tumors of Lewis lung cancer, increase the phagocytosis of macrophages and the proliferation of splenocytes, and increase the contents of tumor necrosis factor-α (TNF-α) and interleukin-2 (IL-2), interleukin-6 (IL-6), and interleukin-12 in serum of mice, which are related to the anti-tumor effect [46]. In addition to its anti-tumor effect, Lilium polysaccharide has a certain synergistic and attenuating effect on tumors combined with other antineoplastic drugs. For example, Lilium polysaccharides can improve the immune function of the body, enhance the anti-tumor effect of chemotherapeutic drugs, and reduce the toxic and side effects of chemotherapy [47]; Lilium polysaccharides can enhance the anti-proliferative effect of metformin on MCF-7 breast cancer cells, which is related to the promotion of apoptosis of MCF-7 breast cancer cells by Lilium polysaccharides [48]. In addition, Lilium polysaccharides can also increase the anti-tumor effects of other traditional Chinese medicines; for example, they can enhance the inhibitory effect of genistein on MCF-7 cell proliferation [49].

#### 3.1.2. Anti-Tumor Effect of Saponins

Some studies have confirmed that total saponins of Lilium can inhibit the proliferation, migration, and invasion of lung cancer cells [50]. In addition, studies have confirmed that Lilium steroidal sapogenin S3 and its derivatives can significantly inhibit the proliferation of HeLa in vitro and inhibit the proliferation of pancreatic cancer, osteosarcoma, human gastric cancer cells (HGC-27), and pheochromocytoma to a certain extent. Moreover, they can enhance anti-tumor ability by adjusting the structure of Lilium steroidal sapogenin S3 [51,52,53].

#### 3.1.3. Anti-Tumor Effect of Alkaloids

The alkaloids in Lilium are also the key components of Lilium’s anti-tumor effect. Some studies have confirmed the anti-tumor effects of alkaloids in different varieties of Lilium. One result shows that alkaloids, saponins, and methanol extracts can inhibit the proliferation of human lung cancer cells in vitro, and the inhibitory effect of alkaloids is the strongest [54]. In addition, some studies have confirmed that Lilium alkaloids can significantly inhibit the proliferation of human gastric cancer cells in vitro, which may be related to the fact that alkaloids can make cancer cells stagnate in the G2/M phase, up-regulate the expression of caspase-3 protease, and induce apoptosis [55]. For example, colchicine can inhibit the proliferation of tumor cells by inhibiting the mitosis of tumor cells, blocking the cell cycle and thus inhibiting the proliferation of tumor cells [55].

#### 3.1.4. Anti-Tumor Effect of Flavonoids

The molecular pathways of the anti-tumor activity of flavonoids are diverse, involving almost all stages of the occurrence and development of cancer. In the stage of tumorigenesis, flavonoids can inhibit DNA damage and promote the recovery of DNA damage [56]. In the stage of tumor development, flavonoids can induce cell cycle arrest and apoptosis and inhibit tumor invasion and metastasis as well as tumor angiogenesis [57].

#### 3.1.5. Anti-Tumor Effect of Emodin

Emodin exerts its anti-tumor effect by inducing apoptosis, inhibiting tumor invasion and metastasis, and enhancing the sensitivity of antineoplastic drugs [58].

### 3.2. Anti-Depressant

Depression is a recurrent and life-threatening mental disorder. A series of studies have shown that the occurrence of depression is related to neurotransmitters such as norepinephrine, 5-hydroxytryptamine, and dopamine and cytokines such as interleukin-1β, IL-2, and IL-6 [59]. Lilium saponins are the main effective components of anti-depressants, and they can significantly reduce the immobility time of tail suspension and forced swimming in mice, which are commonly used to detect depression in mice. Studies have confirmed that Lilium saponins can reduce the contents of serum cortisol and corticotropin in depressed rats, inhibit the expression of CRF mRNA in the hypothalamus, alleviate the decline of monoamine neurotransmitters in depressed rats, and increase the contents of dopamine and serotonin in the cerebral cortex [60]. This suggests that the possible anti-depressant mechanisms of Lilium saponins are as follows: (1) increasing the contents of dopamine and serotonin in the brain of depression model rats and improving the deficiency of monoamine neurotransmitters; (2) increasing the expression of glucocorticoid and corticotropin in the hippocampus and inhibiting the hyperactive hypothalamus–pituitary–adrenal axis; and (3) reducing the content of blood cortisol and corticotropin as well as the expression of hypothalamic corticotropin-releasing factor. In addition, compared with fine leaf Lilium, studies have confirmed that the anti-depressant effect of total saponins is more obvious, and it can improve gastrointestinal discomfort caused by depression by increasing the content of gastrin and substance P in the gastric antrum. A study showed that the anti-depressant mechanism of total saponins of Rhizoma Corydalis may be related to the interaction between the serotonergic nervous system and the brain–gut axis, and the specific mechanism needs further study [61]. In addition, for the rat model of depression with irritable bowel syndrome, Lilium can also relieve the symptoms of depression and visceral hyperalgesia [62].

### 3.3. Anti-Diabetic

Studies have shown that Lilium saponins and Lilium polysaccharides may be potential hypoglycemic ingredients in Lilium. The results of in vitro experiments showed that Lilium saponins could increase glucose consumption in HepG2 cells and 3T3-L1 adipocytes. In addition to Lilium saponins, in vitro studies showed that Lilium polysaccharides of 5.6~16.7 mmol/L could promote islet β-cell proliferation and insulin secretion but had no significant inhibitory effect on α-glucosidase activity [63]. Lilium polysaccharides can reduce the blood glucose concentration of alloxan-induced hyperglycemic mice in a dose-dependent manner [64]. Moreover, some studies have found that Lilium polysaccharides can increase the activities of hexokinase, succinate dehydrogenase, and total superoxide dismutase and reduce the level of fasting blood glucose in type I diabetic rats by reducing the content of MDA [65]. In addition, solanum lanceolate polysaccharide can slow down the weight loss of mice induced by streptozotocin and significantly reduce the level of blood glucose, which is related to the repair of islet injury caused by streptozotocin and the improvement of structural integrity and function of islet β-cells and tissues [66]. Hence, the anti-diabetic effect of Lilium is related to promoting the repair, proliferation, and secretion of islet β-cells and participating in glucose metabolism.

### 3.4. Anti-Oxidation and Scavenging Free Radicals

Lilium polysaccharides have significant anti-oxidant activity. Some studies have found that Lilium polysaccharides could increase the activities of superoxide dismutase, catalase, and glutathione peroxidase in the blood of aging mice and decrease the levels of oxide peroxides in plasma, liver homogenate, and brain homogenate. In addition to polysaccharides, some studies have confirmed that Lilium saponins have a good scavenging effect on hydroxyl radicals [67]. Phenols [68] and flavonoids [11] also can scavenge 1-diphenyl-2-trinitrophenylhydrazine, superoxide ion, and hydroxyl radical. Lilium polysaccharides, saponins, phenols, and flavonoids are expected to be developed and utilized as natural anti-oxidants. In addition, some studies have found that the extracts of six kinds of Lilium bulbs had strong anti-oxidant activity, and their anti-oxidant activity was positively correlated with the contents of total phenols, total flavonoids, and total flavanols [22]. In addition to Lilium bulbs, polyphenols extracted from Lilium petals also have good anti-oxidant activity. This suggests that Lilium may be a potential source of natural antioxidants in pharmaceutical applications.

### 3.5. Immunomodulation

Some studies have found that Lilium polysaccharides can enhance immunity [69]. An animal experiment confirmed that Lilium polysaccharide can promote the proliferation of splenocytes and increase the level of serum hemolysin in mice by enhancing the phagocytosis of macrophages, the thymus index, and the spleen index. Finally, it enhances the non-specific and specific immune function of normal and immunosuppressive mice [70]. Some studies have shown that Lilium polysaccharides significantly up-regulate the expression of immunoreactive cytokines in RAW264.7 macrophages through toll-like receptor 4-mediated nuclear factor kappa B (NF-κB) signal pathways [71]. Moreover, modification of the structure of Lilium polysaccharides, such as selenite, can significantly improve the immune activity of polysaccharides [72]. In addition, a water-soluble polysaccharide BHP was isolated from Lilium scales, and the results of mouse experiments showed that BHP could effectively regulate immune function [73].

### 3.6. Anti-Inflammatory

The anti-inflammatory effect of Lilium may be related to saponins. Some studies have found that Lilium methanol extract can reduce the swelling of the hindfoot in mice in a dose-dependent manner, and saponins are the main substances of Lilium methanol extract [11]. Then, the anti-inflammatory mechanism of methanol extract of Lilium was explored, and it was found that methanol extract of Lilium can down-regulate the expression of nitric oxide synthase and epoxidase 2 by inhibiting NF-κB activation and nuclear translocation in Raw264.7 cells stimulated by lipopolysaccharide and blocking ERK and JNK signal transduction and c-Jun N-terminal kinase signal transduction [74]. Moreover, the anti-inflammatory effect of Lilium was related to emodin, which can reduce hepatic fibrosis by inhibiting the transforming growth factor-β1 (TGF-β1) signal transduction pathway and protect the liver from pro-inflammatory and pro-oxidative damage caused by a high-fat diet [75]. Recent studies have found that the anti-inflammatory effect of Lilium is also related to phenols and flavonoids in Lilium, and the specific bioactive components, composition, and mechanism need to be further studied. However, current research has not identified a specific pathway by which Lilium components exert their anti-inflammatory effects.

### 3.7. Regulation of Brain–Gut Axis

In recent years, studies have found that there are interconnections and interactions between the gastrointestinal tract and the central nervous system, which is closely related to the “brain–gut axis”. Zhong-jing Zhang used to treat depression and improve the patient’s gastrointestinal symptoms with Lilium. By detecting the level changes of gastrin, vasoactive intestinal peptide, and P-substance in the blood, stomach, and intestine of rats, a study found that Lilium saponins could increase the content of plasma vasoactive intestinal peptide, decrease the content of vasoactive intestinal peptide in colon tissue, and increase the content of gastrin and P-substance in blood and colon tissue, which suggests that Lilium saponins have a certain regulatory effect on the brain–gut axis. In addition, when Lilium was applied to depressed rats, a study found that Lilium could improve the disappearance of pleasure with sugar water in depressed rats and reduce the sense of behavioral despair in swimming rats, which indicated that Lilium could improve the symptoms of depression and improve gastrointestinal discomfort symptoms through the regulation peptides of the brain–gut axis at the same time [61]. However, the specific bioactive components and targets of Lilium that modulate the “brain–gut axis” are not clear.

### 3.8. Anti-Fatigue and Hypoxia Tolerance

Studies have shown that the extract of Lilium polysaccharide has an anti-fatigue effect, which can prolong the weight-bearing swimming time and enhance the anti-fatigue ability in mice [76]. Lilium can improve the ability of the body to endure hypoxia. Some studies have found that Lilium can significantly prolong the hypoxia tolerance time of mice under normal pressure and acute cerebral ischemic hypoxia as well as the survival time of mice with sodium nitrite poisoning [77]. Sichuan Lily can also prolong the time of myocardial hypoxia induced by isoproterenol in mice [78]. The doses of 200 and 400 mg/kg of Lilium polysaccharide can significantly enhance the hypoxia tolerance of mice [79]. In addition, some studies have found that the n-butanol-soluble part of Lilium can significantly prolong the survival time of mice under normobaric hypoxia and prolong the swimming time of mice in ice water baths [80]. In short, Lilium has the effects of anti-fatigue and hypoxia tolerance, but the specific active ingredients are not clear.

### 3.9. Anti-Bacterial

Lilium has anti-fungal and anti-bacterial activity. In the aspect of anti-fungal, some studies have confirmed that the compounds isolated from musk Lilium can inhibit the growth of botrytis cinerea. The specific mechanism may be by inhibiting the metabolic rate of botrytis cinerea [81]. Lilium polysaccharides have a certain inhibitory effect on molds produced by food spoilage and have varying degrees of anti-fungal activity against saccharomyces cerevisiae, aspergillus niger, and rhizopus sinensis [82]. In addition, steroidal alkaloids were found to have weak inhibitory activity against botrytis cinerea [83]. In terms of anti-bacterial activity, alkaloids and polysaccharides in Lilium have anti-bacterial activity. Some studies have found that colchicine extracted from Lilium powder has an obvious inhibitory effect on Escherichia coli, staphylococcus aureus, bacillus subtilis, candida albicans, and sarcina flava [61]. Lilium polysaccharide has an inhibitory effect on staphylococcus aureus to a certain extent [82].

### 3.10. Other Pharmacological Effects

In addition to the above functions, current studies have found that Lilium also has other functions. Research has found that n-butanol extract of steroidal glycosides has a hepatoprotective effect [84]. The flavonoids in the water extract and methanol extract of Lilium tenuifolia also have a cholagogic effect, and the water extract can increase the bile flow in a dose-dependent manner [85]. Flavonoids isolated from Lilium can reduce the cytotoxicity, genotoxicity, and teratogenicity of bleomycin [86]. Moreover, some studies have found that Lilium extract can inhibit the synthesis of melanin, which suggests that Lilium may have a whitening effect [87,88]. In addition, some steroidal alkaloids and furosteroid soaps can promote wound healing [89]. The anti-viral activity of seven phenolic acid glycerides isolated from the bulb was tested, and results showed that compound 42 and compound 48 could significantly inhibit the activity of respiratory syncytial virus [90]. In addition, research found that saponins showed strong inhibitory effects on Na^+^/K^+^ ATP enzyme activity [91].

### 3.11. Security

Lilium, especially Lanzhou lily, has been used as food on the table for a long time and for people’s health. Lilium compound preparation has been used since ancient China and has a good security. For example, in a clinical study, 150 patients with advanced lung cancer complicated with pneumonia were treated with Baihe Gujin decoction (BGD) (a traditional Chinese medicine). No obvious side effects were found after 2 weeks of treatment [92]. A study found that some species of the genus Lilium are nephrotoxic to cats and dogs, who developed vomiting and gastrointestinal symptoms after eating lilies. However, for rabbits and mice, nephrotoxicity did not occur even when fed up to one and a half times their body weight [93]. Some research has indicated that Lilium may be toxic to humans. A case report showed that an 87-year-old patient with dementia developed acute symptoms of poisoning similar to digitalis toxicity after consuming the stems and leaves of wild lily of the valley [28]. Another case report showed that five young people developed acute neurotoxic symptoms after consuming Angel trumpet lily [94]. In addition to acute poisoning, a case report showed that Lilium may cause anaphylaxis [95]. Due to the frequent combined use with other traditional Chinese medicines, no cases of chronic poisoning caused by Lilium have been reported. However, modern medical research has revealed that Lilium contains colchicine, which may produce toxic effects when taken over a long period. Therefore, studies are urgently needed to explore the possibility of chronic poisoning caused by Lilium [96,97]. Standardizing the dosage of Lilium is essential to mitigate its side effects. In China, the clinical dosage of Chinese medicines is based on established clinical standards and quality control results of the active ingredients. According to the Chinese Pharmacopoeia, Lilium is used for the treatment of palpitations, insomnia and hemoptysis, with a recommended dosage range of 6–12 g. In addition, a study synthesizing ancient medical texts and modern clinical practice established the current dose range for Lilium as 5.55 to 60 g for oral use and up to 65 g for external use [98]. This suggests that Lilium is not significantly toxic to humans in this dose range. However, potential side effects are still a concern with long-term use of Lilium. Lilium is cool-natured and should be discontinued if symptoms such as nausea, vomiting, abdominal pain, or diarrhea occur [96]. More studies on the toxicity of Lilium are needed in the future to provide a reliable basis for clinical drug safety.

### 3.12. Lilium Acts through the Interaction between Multiple Organs

It is complex for Lilium active ingredients to play a role in the treatment of diseases, and it is not clear through what mechanism Lilium plays a therapeutic role. Lilium, as a nutraceutical, is very likely to change the microecosystem of the body after eating [99]. Lilium can affect the distribution of intestinal microflora after entering the intestinal tract and plays a therapeutic role through the interaction of multiple organs. For example, Lanzhou lily has a high content of polysaccharides. Although the human body cannot digest plant polysaccharides other than starch directly, these non-starch plant polysaccharides can be used as probiotics to maintain gastrointestinal health by inhibiting pathogens and stimulating the immune system. Bioactive substances are produced after being absorbed by intestinal flora, thus regulating the structure of intestinal microorganisms and being transported to multiple organs of the body to play a role [100]. Several studies have found that polysaccharides can regulate intestinal microecological balance and improve cognitive impairment through the brain–gut axis [101]. In addition, because of its complex active components, traditional Chinese medicine often plays a therapeutic role in multiple targets and pathways of the human body [102]. In recent years, integrative pharmacology has become a key paradigm for the modernization and combinatorial drug discovery of traditional Chinese medicine. This is to establish a cross-discipline of in vivo and in vitro correlation between the absorption, distribution, metabolism, excretion, and pharmacokinetics spectrum of traditional Chinese medicine and the molecular network of diseases by integrating multi-disciplinary and multi-stage knowledge [103]. With the progress of technical means, especially the development of integrative pharmacology, more research is needed on how Lilium components play a therapeutic role through the interaction of organs.

The bioactive components of Lilium have a wide range of plant pharmacological effects, but many functions are unknown and need to be discovered. The components and contents of Lilium with different geographical distributions are different, which determines that they play different roles. Moreover, the mechanism of Lilium in treating diseases is also very complex and has not been revealed clearly (Figure 3). In addition, there are few studies on the pharmacology and toxicity of Lilium bioactive components at present. Future studies should strengthen the toxicity of Lilium bioactive components to provide a meaningful reference for the safe and effective use of drugs.

## 4. Baihe: The Application of Traditional Chinese Medicine

The compound preparation based on Baihe has a wide range of pharmacological effects, its clinical application is extensive, and the curative effect is reliable, such as BDD, BZD, and BGD (Figure 4).

### 4.1. Baihe Dihuang Decoction (BDD)

BDD is a well-known and classical Chinese prescription from Jin Gui Yao Lve (Eastern Han Dynasty, AD 25–220). The decoction is the aqueous solution of L. brownii F.E.Br. ex Miellez bulbs and Rehmannia glutinosa roots. It is reported that BDD is mainly used in the treatment of peri-menopausal syndrome, insomnia, and lung diseases [104]. It has the function of nourishing yin, clearing heat, and moistening the heart and lungs. In modern times, it is often used for diseases such as lung disease, peri-menopausal syndrome, depression, and insomnia [105,106]. Modern research shows that BDD contains various bioactive components such as lily polysaccharides, steroidal saponins, adenosine, glutamic acid, and alkaloids, and the pharmacological effects of BBD are related to these bioactive components [107]. BDD can treat lung cancer. Research has found that chemotherapy combined with BDD can improve the chemotherapeutic effect of small-cell lung cancer. It can also improve the adverse reactions of the digestive tract caused by chemotherapy and improve the nutritional status of patients undergoing chemotherapy [108]. The modified BDD has a therapeutic effect on post-stroke depression, which may be related to up-regulating the expression of genes related to neuronal proliferation, development, differentiation, and migration and promoting the repair of neural structure and function [109]. BDD may improve depression-like behavior by regulating the metabolism of amino acids, steroids, and glycerophospholipids in depressed rats [110]. The modified BDD combined with acupoint acupuncture is effective in the treatment of uremic hemodialysis sleep disorder, which can effectively improve sleep quality and anxiety symptoms and not affect the effect of hemodialysis [111]. A study has found that BDD can restore the excessive activation of inflammation and the imbalance of neurotransmitters induced by corticosterone by increasing the level of anti-inflammatory factor interleukin 10, γ-aminobutyric acid, and serotonin [112]. The modified BDD combined with flupentixine tablets is effective in the treatment of peri-menopausal depression, which can improve peri-menopausal symptoms and depressive state, regulate the level of serum neurotransmitters, and reduce the incidence of adverse reactions [113]. BDD combined with trazodone hydrochloride is effective in the treatment of insomnia associated with yin deficiency depression with few adverse reactions [114].

### 4.2. Baihe Zhimu Decoction (BZD)

BZD is a well-known and classical Chinese prescription from Jin Gui Yao Lve, which was composed of L. brownii F.E.Br. ex Miellez bulbs and Anemarrhena anemarrhena roots and used in the clinical treatment of depression [115]. It was originally used to treat patients with Lilium disease after sweating. At present, this prescription is effective in the treatment of depression caused by the internal heat of yin deficiency, unclear residual heat, depression, and resolving fire [116]. BZD combined with Tong Du Tiao Shen acupuncture is effective in the treatment of post-stroke depression, which can reduce neurological impairment, improve patients’ depressive state, and improve their daily life [117]. BZD combined with Morita therapy can improve the symptoms, quality of life, and activities of daily life in patients with first-episode depression, which may be related to the decrease of serum brain-derived neurotrophic factor and the increase of dihydroxyphenylacetic acid level [118]. BZD combined with Yu Yuan acupoint acupuncture is effective in the treatment of senile post-stroke depression, which can effectively improve the condition of the patients and is beneficial to the prognosis [119]. BZD primarily contains active ingredients such as polysaccharides, alkaloids, organic acids, sterols, flavonoids, and saponins. A study based on the integrative pharmacology-based research platform of traditional Chinese medicine found that BZD’s treatment of depression is related to these bioactive components targeting genes or proteins such as GABRA1, NFKB1, AR, SMC1A, PIK3CA, NR3C1, and DRD2, affecting pathways such as gamma-aminobutyric acid, chloride ion transmembrane transport, postsynaptic membrane potential, negative regulation of neuronal apoptosis processes, and signal transduction effects [120].

### 4.3. Baihe Gujin Decoction (BGD)

BGD is a common prescription for treating cough due to a deficiency of lung and kidney yin. It can be used for long coughs with lung deficiency and dry cough with less phlegm. BGD combined with fermented Dong Chong Xia Cao powder in the treatment of lung cancer with lung and kidney yin deficiency can not only alleviate the clinical symptoms and improve the therapeutic effect of tumors but also enhance the immune function of patients [121]. The anti-tumor activity of BGD is associated with its bioactive compounds kaempferol, quercetin, isorhamnetin, glycyrrhizin, and β-sitosterol [122]. The modified BGD combined with acupoint needle embedding is effective in the treatment of chronic obstructive pulmonary disease, which can effectively improve clinical symptoms and dyspnea and improve pulmonary function [123]. BGD combined with anti-tuberculosis drugs is effective in the treatment of pulmonary tuberculosis, which can effectively improve the level of T lymphocyte subsets and correct malnutrition [124]. The modified BGD combined with Western medicine makes sputum bacteria-negative, focuses on absorption, and has less adverse reactions, which helps improve the clinical symptoms of newly treated smear-positive pulmonary tuberculosis patients [125]. BGD combined with Fu Fang Chuan Xin Lian is effective in the treatment of bronchiectasis of yin deficiency and lung heat, which can effectively improve the clinical symptoms and pulmonary function of the patients [126]. BGD can effectively treat ventilator-associated pneumonia of lung and kidney yin deficiency type, and the therapeutic effect is better than that of the control group treated with Western medicine alone. It can reduce the inflammatory index, acute physiology and chronic health evaluation II score, cough and sputum, shortness of breath, night sweats, and other symptoms [127]. BGD can effectively treat patients with pulmonary tuberculosis complicated with diabetes mellitus [128]. The curative effect of BGD combined with Zhi Ke San in the treatment of lung cancer cough (lung yin deficiency syndrome) is better than that of codeine phosphate tablets [129]. In addition, BGD has a significant hemostatic effect, and its hemostatic mechanism is related to activating the endogenous coagulation pathway, activating the fibrinolytic system, and promoting platelet aggregation [130].

### 4.4. Other Compound Preparations Based on Lilium

In addition to classic BDD, BZD, and BGD, other compound preparations based on Lilium are also widely used in clinics. For example, Ganwei Baihe decoction can reduce the expression of Beclin-1, light chain 3B (LC3B), and LC3B-II/LC3B-I; increase the expression of p62; and alleviate gastric mucosal injury in SU rats by inhibiting overactivated autophagy [131]. Ganwei Baihe decoction combined with Vonolasone can effectively relieve the clinical symptoms of patients with reflux esophagitis [132]. Baihe Runjing decoction combined with artificial tears can increase tear secretion, improve corneal fluorescein sodium staining, prolong tear film rupture time, increase lacrimal river height, and improve patients’ subjective symptoms [133]. Baihe Danshen Yin can treat diabetic ketoacidosis, diabetic gastroparesis, diabetes complicated with hypertension, and other diseases [134]. Baihe Wuyao decoction (BWD) can improve the symptoms of liver injury caused by type 1 diabetes through anti-inflammation, anti-oxidation, and anti-apoptosis, promoting cell proliferation and improving the insulin signal pathway [135]. BWD decoction relieves chronic liver injury induced by CCl4 and liver fibrosis in mice via blocking TGF-β1/Sma-and Mad-Related Protein-2/3 signaling, anti-inflammation, and anti-oxidation effects [136]. Baihe adenocarcinoma decoction combined with gefitinib in the treatment of advanced lung adenocarcinoma with internal heat due to yin deficiency can reduce the occurrence of side effects and prolong the progression-free survival time of the patients [137]. Baihe Dihuang Pingkang decoction based on conventional Western medicine treatment in the treatment of hyperthyroidism can improve the curative effect and reduce the adverse reactions and recurrence rate [138]. Baihe Shugan Anshen decoction maximizes the effect of anti-depressants by regulating the hypothalamic pituitary adrenal axis and the level of monoamine neurotransmitters in the brain [139]. Baihe Jizihuang Tang (BHT) has a certain anti-depressant effect, which may play a role through brain-derived neurotrophic factor/tropomyosin receptor kinase B and its downstream phosphatidylinositol 3kinase/AKT/mammalian target of rapamycin signal pathway [140]. Baihe Shuxin decoction can reduce the levels of serum creatine kinase (CK), CK-MB, and lactate dehydrogenase in patients with angina pectoris after percutaneous coronary intervention, which is beneficial in improving the symptoms of angina pectoris [141]. Shuanghua Baihe Tablet can reduce the incidence of severe mucositis caused by radiotherapy for head and neck tumors [142]. Moreover, an ancient Persian remedy (a mixture of white Lilium in sesame oil) can relieve the symptoms of patients with chronic lower back pain [143].

Baihe-based compound preparation is modern medicine based on ancient prescriptions. Compound preparation has a better therapeutic effect through the combination of traditional Chinese and Western medicine to add and subtract some drugs, and this kind of compound preparation is being used more and more in clinical application. The components of Lilium compound prescriptions interact with each other in the treatment of diseases; some promote each other to enhance the therapeutic effect, and some meet each other to reduce side effects. More in-depth study on the compatibility of traditional Chinese medicine in Lilium compound preparation is needed in the future.

## 5. Summary and Prospect

Lilium has rich pharmacological activities and is one of the classical homologous varieties of medicine and food. The main bioactive components of Lilium are polysaccharides, steroidal saponins, and phenols, and modern pharmacological studies have gradually explained and excavated the traditional efficacy of Lilium. However, there are still many problems in the current research: (1) The study of the efficacy of Lilium remains in the effectiveness of total components, but the study of the relationship between components and activity is not in-depth. Some of the results suggest that there are certain rules between the structural differences of the steroidal saponin, anti-tumor, and anti-depressant activity of Lilium, and their specific relationship needs to be further studied. Moreover, there are few studies on the pharmacology and toxicity of Lilium components. It is recommended that the toxicity studies of Lilium components should be strengthened to provide useful references for safe and effective medication. (2) There are many varieties of Lilium, and the content of components is greatly affected by the environment. The quality of the Lilium is controlled only by character and TLC identification, and there is a lack of quantitative standards of effective components. Therefore, improving the quality evaluation standard of Lilium helps control the quality of Lilium. Some scholars have preliminarily established the quality standard of Lilium by determining the content of steroidal saponins by UV-vis spectrophotometry, high-performance liquid chromatography, and fingerprint technology, but the related research needs to be further promoted. (3) At present, the development and application of Lilium are relatively few, mostly remaining only at the stage of crude food, and the research and development of drugs for their anti-tumor, anti-depressant, anti-oxidant, and anti-diabetic effects are not yet in-depth. (4) The properties of Lilium from different regions determine the differences in their compositions, the differences in compositions determine the differences in components, and the differences in components determine the differences in medicinal values. At present, the research and excavation of the properties of Lilium from different regions are still insufficient, which is one of the important directions for future research. (5) The effects of different components of Lilium are different, and the components with high Lilium content will not necessarily play a role. The study of components with small Lilium content but with important pharmacological effects is also the direction of future research. (6) There are many common and similar components in the ingredients of plants. How to explore the special ingredients and use them in combination with modern medicine is an urgent problem to be solved.

In the future, the following research of Lilium should be strengthened: (1) Explore whether Lilium has chronic toxic side effects on humans when used as a whole or as a single component. (2) Lucubrate the pharmacological effects of Lilium, especially through clinical experiments to observe whether it has anti-tumor, anti-oxidant, and immunity-enhancing effects, and even explore new pharmacological effects of Lilium. (3) There are many reports on the effective components and pharmacological mechanisms of compound preparations, but there are few basic studies based on integrity. Research should be strengthened on the content of bioactive components in traditional prescriptions and how these bioactive components are metabolized and exert their effects in the body so as to provide modern research support for the clinical application and in-depth development of traditional Chinese medicine. (4) Lucubrate the interaction between Lilium and other Chinese herbal medicines, explore more possible compatibility, and provide more treatment options for clinical practice.

Many people are involved in the research of Lilium, and there are more and more cross-integrations in different fields, such as ecology, agronomy, chemistry, and medicine. In recent years, the development of single-cell technology and metabonomic technology has provided convenience for the research of Lilium. It is believed that with the in-depth study of Lilium, the application prospects of Lilium in the fields of food and medicine will be broader. Starting from the geographical location, determine the active components of food and drug substances, then explore the diseases and related mechanisms of these components, which is the general law of the study of food and drug substances.

## Figures and Tables

**Figure 1 pharmaceuticals-17-01242-f001:**
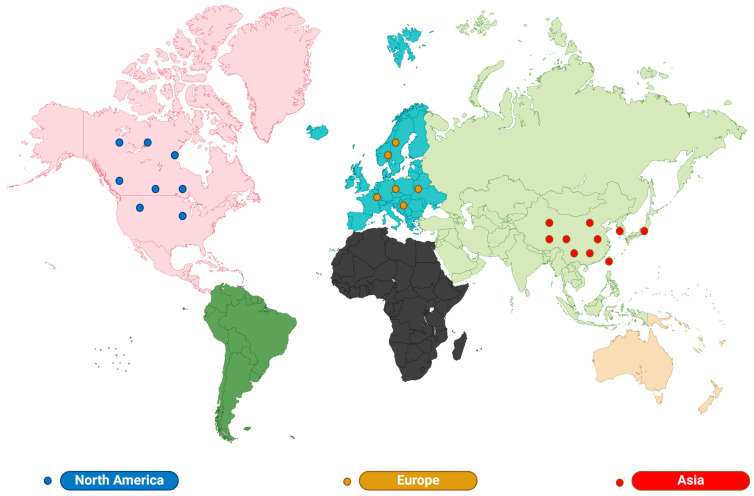
The world distribution of Lilium. Lilium is mainly distributed in the Northern Hemisphere within temperate regions, such as Asia, Europe, and North America.

**Figure 2 pharmaceuticals-17-01242-f002:**
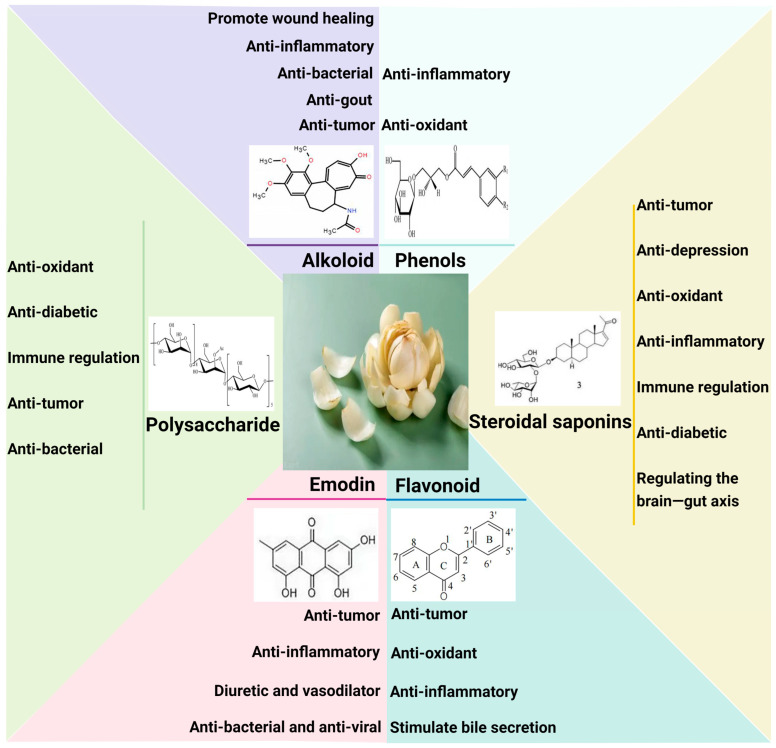
Pharmacological effects of Lilium bioactive components. The bioactive components of Lilium include steroidal saponins, polysaccharides, phenols, flavonoids, and alkaloids. These bioactive components have therapeutic effects on a variety of diseases, such as anti-tumor, anti-depressant, anti-diabetic, anti-oxidation and scavenging free radicals, immunomodulation, anti-inflammatory, regulation of brain–gut axis, anti-fatigue and hypoxia tolerance, and anti-bacterial.

**Figure 3 pharmaceuticals-17-01242-f003:**
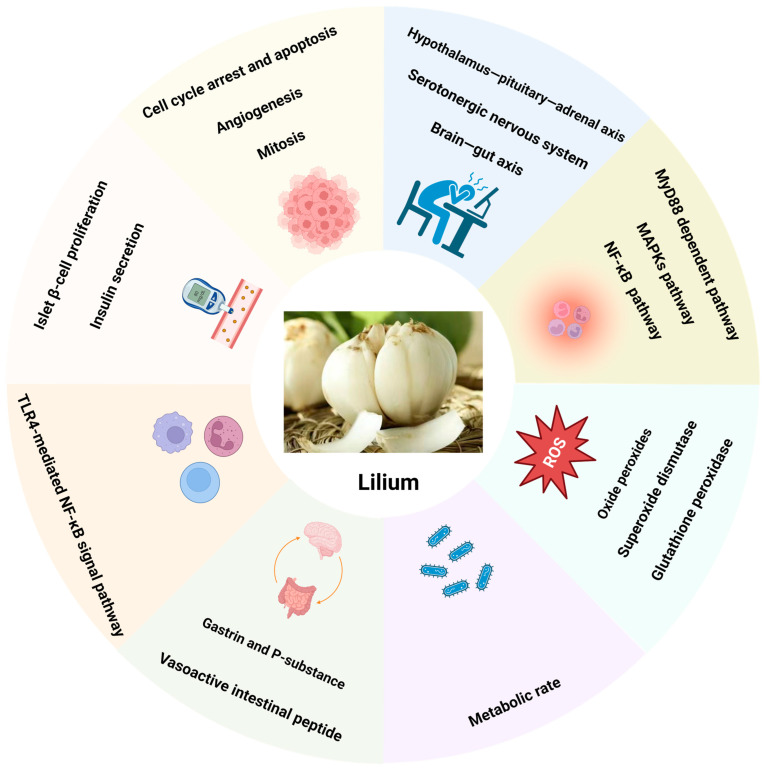
The mechanism of Lilium in treating disease. The mechanism of Lilium in treating diseases is very complex and has not been established clearly.

**Figure 4 pharmaceuticals-17-01242-f004:**
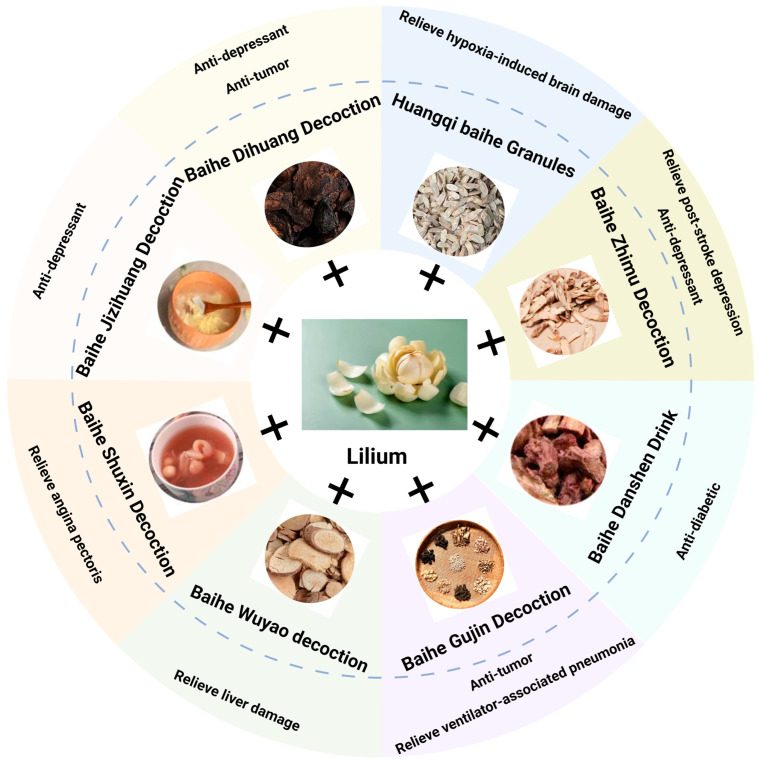
Pharmacological effects of compound preparations based on Lilium. The compound preparation of traditional Chinese medicine with Lilium as the main drug has a wide range of pharmacological effects, its clinical application is extensive, and the curative effect is reliable, such as BDD, BZD, and BGD. These compound preparations take Lilium as the main medicine, together with other traditional Chinese medicine ingredients, and play a role in the treatment of diseases. The use of Lilium with these traditional Chinese medicine ingredients will increase the efficacy of single drugs or reduce side effects, making Lilium safer and more effective in the treatment of diseases.

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
