# Peer review of "Lilium brownii/Baihe as Nutraceuticals: Insights into Its Composition and Therapeutic Properties"

_pharmaceuticals, 2024, doi:10.3390/ph17091242_

Round 1

Reviewer 1 Report

Comments and Suggestions for Authors

1. The crucial point to emphasize in this article is the need to distinguish between food and medicine, as they are fundamentally different. The concept of functional food is unique and should not be confused with medicinal food. Functional foods are those typically consumed as part of a daily diet with the aim of preventing disease, whereas medicine is taken in specific doses with the purpose of curing illnesses. If the author wishes to bridge the gap between food and medicine, the term "nutraceuticals" is more appropriate. Nutraceuticals are compounds or components in food that offer health benefits. They can be incorporated into food to make it functional or used as supplements or medicine.

2. Please change whole story in the manuscript.

Author Response

Comments 1:1. The crucial point to emphasize in this article is the need to distinguish between food and medicine, as they are fundamentally different. The concept of functional food is unique and should not be confused with medicinal food. Functional foods are those typically consumed as part of a daily diet with the aim of preventing disease, whereas medicine is taken in specific doses with the purpose of curing illnesses. If the author wishes to bridge the gap between food and medicine, the term "nutraceuticals" is more appropriate. Nutraceuticals are compounds or components in food that offer health benefits. They can be incorporated into food to make it functional or used as supplements or medicine.

Response 1:Thank you for your comments. Based on your suggestions, we examined the term "nutraceuticals". Nutraceuticals or natural products are commonly called medical foods. Nutraceuticals are compounds or components in food that offer health benefits. We agree with you that "nutraceuticals" better conveys the core meaning of this review, so we have changed "Functional Medicinal Food" to "nutraceuticals". We also reviewed relevant literature and found that "Functional foods"and" nutraceuticals "are difference (Mao, X. Y., Jin, M. Z., Chen, J. F., Zhou, H. H., & Jin, W. L. (2018). Live or let die: Neuroprotective and anti-cancer effects of nutraceutical antioxidants. Pharmacology & therapeutics, 183, 137–151. https://doi.org/10.1016/j.pharmthera.2017.10.012). Thank you very much for your suggestion.

Comments 2: Please change whole story in the manuscript.

Response 2:

Thank you for your comments. Based on your suggestions, we have changed "Functional Medicinal Food" to "nutraceuticals" in the whole story. Our modifications are as below:

In the abstract and introduction section, we introduce "nutraceuticals" and define Lilium brownii / Baihe as a nutraceutical. Our modifications are as below:

Line 13-14: Nutraceuticals are compounds or components in food that offer health benefits. They can be incorporated into food to make it functional or used as supplements or medicine. Lilium brownie/Baihe is one of the classic nutraceuticals.

Line 33-36: Nutraceuticals or natural products are commonly called medical foods. The continuous intake of nutraceuticals has a beneficial effect on human health and can improve symptoms of certain diseases, such as cardiovascular diseases, diabetes, neurological diseases, and cancers.

Reviewer 2 Report

Comments and Suggestions for Authors

The paper focuses on the bioactive components and therapeutic properties of Lilium brownii/Baihe, a classic example of functional medicinal food in traditional Chinese medicine. The study provides a comprehensive analysis of its chemical composition, including saponins, polysaccharides, flavonoids, and alkaloids, and explores its pharmacological effects, such as anti-tumor, anti-depressant, anti-diabetic, and immunomodulatory activities. The paper also discusses the historical use of Lilium brownii in traditional Chinese medicine, highlighting its role in treating respiratory and mental health disorders. The following issues should be addressed to provide a more comprehensive and authoritative review article:.

1. While the paper acknowledges that regional differences affect the composition of Lilium brownii, it does not fully explore how these variations influence its therapeutic efficacy. The paper should include a more thorough investigation into how geographical and environmental factors influence the chemical composition and therapeutic potency of Lilium brownii. Comparative studies across different regions could provide valuable insights into optimizing cultivation practices for enhanced medicinal properties.

2. The paper briefly mentions the safety of Lilium brownii, especially its nephrotoxicity in animals, but does not provide a comprehensive evaluation of its toxicity in humans. A more detailed toxicological assessment should be conducted and included, with a focus on both acute and chronic toxicity in humans.

3. Can the authors add further research to determine safe dosage ranges and identify any potential long-term side effects? Reviewing existing safety data from traditional use could also help reduce concerns.

4. Can the application of Baihe in traditional Chinese medicine be linked to its multiple bioactive components? Explain the mechanism of its important role in traditional Chinese medicine to help people who are not familiar with traditional Chinese medicine better understand its pharmacological effects.

5. The paper should clearly identify gaps in current research and suggest specific areas where further studies are needed in the conclusion /discussion. This could include unexplored pharmacological effects, potential interactions with other drugs, or under-researched therapeutic applications.

Comments on the Quality of English Language

Minor editing of the English language is required.

Author Response

一、Comments 1:While the paper acknowledges that regional differences affect the composition of Lilium brownii, it does not fully explore how these variations influence its therapeutic efficacy. The paper should include a more thorough investigation into how geographical and environmental factors influence the chemical composition and therapeutic potency of Lilium brownii. Comparative studies across different regions could provide valuable insights into optimizing cultivation practices for enhanced medicinal properties.

Response 1:Thank you for your comments. Based on your suggestions, we have found studies that document variations in Lilium composition in different geographic locations, revealing why some are edible and others have medicinal value. The bioactive components in medicinal preparations of Lilium are related to climatic factors. Even the same Lilium grown in different regions has significant differences in the content of its bioactive components. In addition, Lilium from Europe, Asia, and North America are used for the treatment of a variety of diseases, which may be related to differences in the bioactive components contained in Lilium from different geographical regions. Our modifications are as below:

Line 135-162:Furthermore, there are many varieties of Lilium, and Lilium grown in different geographic regions have different bioactive components, which may affect their medicinal properties. The main bioactive components of Lilium are saponins and polysaccharides. The planting range of Lilium is wide, and the quality of Lilium varies greatly among different areas. The growth of lilium is significantly affected by specific ecological and climatic conditions, which are directly related to the quality of the lilium [23]. A study found that temperature and rainfall are the key climate factors for the quality formation of Lilium, and high-temperature climates promote the accumulation of total polysaccharides in Lilium [24]. This geographical difference is closely related to the efficacy of the Lilium. Firstly, in terms of taste, Lilium lancifolium Thunb in Longshan Hunan, also known as "Longshan lily", has a bitter flavour. The high saponin content of Longshan lily makes it a classic medicinal Lilium [25]. However, Lanzhou lily from Gansu Province are sweet and tasty and have a high polysaccharide content, making them a local speciality food [26]. In addition, there are significant differences in the bioactive components of a lily species in different geographical regions. A study analysed the polysaccharide content in Lanzhou lily from six major origins in and around Lanzhou and found that the polysaccharide content of Lanzhou lily from different origins varied considerably [27]. Secondly, the medicinal properties of lilium vary from one geographical location to another. Most Lilium are usually non-toxic and have therapeutic properties for a wide range of ailments. However, Lily of the valley, which grows throughout Europe, North America, and Asia, can cause acute poisoning when consumed [28]. In addition, Lilium is mainly used to treat mastitis, liver disease, and herpeszoster in Europe [29-32]. In Asia, Lilium is mainly used to treat lung diseases [33-36]. Whereas in North America, Lilium is mainly used for food [37,38]. This may be related to the bioactive components contained in lilium from different regions. No studies have yet reported on the specific differences in efficacy of Lilium across different geographical locations. Future research in this area will provide scientific basis for rational planning and cultivation of Lilium medicinal materials, as well as sustainable utilization of resources.

Comments 2: The paper briefly mentions the safety of Lilium brownii, especially its nephrotoxicity in animals, but does not provide a comprehensive evaluation of its toxicity in humans. A more detailed toxicological assessment should be conducted and included, with a focus on both acute and chronic toxicity in humans.

Response 2:Thank you for your comments. Based on your suggestions, we reviewed the literature and found that certain species of Lilium may cause poisoning in humans. Our modifications are as below:

Line 447-457:Some research has indicated that Lilium may be toxic to humans. A case report showed that an 87-year-old patient with dementia developed acute symptoms of poisoning similar to digitalis toxicity after consuming the stems and leaves of wild lily of the valley [28]. Another case report showed that five young people developed acute neurotoxic symptoms after consuming Angel trumpet lily [94]. In addition to acute poisoning, a case report showed that Lilium may cause anaphylaxis [95]. Due to the frequent combined use with other traditional Chinese medicines, no cases of chronic poisoning caused by Lilium have been reported. However, modern medical research has revealed that Lilium contains colchicine, which may produce toxic effects when taken over a long period. Therefore, studies are urgently needed to explore the possibility of chronic poisoning caused by Lilium [96,97].

Comments 3: Can the authors add further research to determine safe dosage ranges and identify any potential long-term side effects? Reviewing existing safety data from traditional use could also help reduce concerns.

Response 3:Thank you for your comments. Based on your suggestions, we reviewed relevant literature and identified the safe dosage ranges of Lilium from traditional use. Our modifications are as below:

Line 457-468:So standardising the dosage of Lilium is essential to mitigate its side effects. In China, the clinical dosage of Chinese medicines is based on established clinical standards and quality control results of the active ingredients. According to the Chinese Pharmacopoeia, Liliumof is used for the treatment of palpitations, insomnia and haemoptysis, with a recommended dosage range of 6-12 grams. In addition, a study synthesising ancient medical texts and modern clinical practice established the current dose range for Lilium: 5.55 to 60 grams for oral use and up to 65 grams for external use [98]. This suggests that Lilium is not significantly toxic to humans in this dose range. However, potential side effects are still a concern with long-term use of Lilium. Lilium is cool-natured and should be discontinued if symptoms such as nausea, vomiting, abdominal pain, or diarrhoea occur [96]. More studies on the toxicity of Lilium are needed in the future to provide a reliable basis for clinical drug safety.

Comments 4: Can the application of Baihe in traditional Chinese medicine be linked to its multiple bioactive components? Explain the mechanism of its important role in traditional Chinese medicine to help people who are not familiar with traditional Chinese medicine better understand its pharmacological effects.

Response 4:Thank you for your comments. Based on your suggestions, we reviewed relevant literature and found the application of Baihe in traditional Chinese medicine is linked to its multiple bioactive components. Our modifications are as below:

Line 526-529:Modern research shows that BDD contains various bioactive components such as lily polysaccharides, steroidal saponins, adenosine, glutamic acid, and alkaloids, and the pharmacological effects of BBD are related to these bioactive components [107].

Line 563-570:BZD primarily contains active ingredients such as polysaccharides, alkaloids, organic acids, sterols, flavonoids, and saponins. A study based on the integrative pharmacology-based research platform of traditional Chinese medicine found that BZD's treatment of depression is related to these bioactive components targeting genes or proteins such as GABRA1, NFKB1, AR, SMC1A, PIK3CA, NR3C1, and DRD2, affecting pathways such as gamma-aminobutyric acid, chloride ion transmembrane transport, postsynaptic membrane potential, negative regulation of neuronal apoptosis processes, and signal transduction effects [120].

Line 577-578:The anti-tumor activity of BGD is associated with its bioactive compounds kaempferol, quercetin, isorhamnetin, glycyrrhizin, and β-sitosterol [122].

Comments 5: The paper should clearly identify gaps in current research and suggest specific areas where further studies are needed in the conclusion /discussion. This could include unexplored pharmacological effects, potential interactions with other drugs, or under-researched therapeutic applications.

Response 5:Thank you for your comments. Based on your suggestions, we have identified specific areas for further research in the conclusions section. Our modifications are as below:

Line 676-688:In the future, the following research of Lilium should be strengthened: 1. Explore whether Lilium has chronic toxic side effects on human when used as a whole or as a single component; 2. Lucubrate the pharmacological effects of Lilium, especially through clinical experiments to observe whether it has anti-tumor, anti-oxidant, and immunity-enhancing effects, and even explore new pharmacological effects of Lilium; 3. There are many reports on the effective components and pharmacological mechanisms of compound preparations, but there are few basic studies based on integrity. Research should be strengthened on the content of bioactive components in traditional prescriptions, and how these bioactive components are metabolized and exert their effects in the body, so as to provide modern research support for the clinical application and in-depth development for traditional Chinese medicine; 4. Lucubrate the interaction between Lilium and other Chinese herbal medicines, explore more possible compatibility, and provide more treatment options for clinical practice.

二、Minor editing of the English language is required.

R: Based on your suggestions, we have carefully read and thoroughly revised the entire manuscript, correcting a number of grammatical errors.

We sincerely acknowledge for your helpful advice on this manuscript!

Round 2

Reviewer 1 Report

Comments and Suggestions for Authors

The author has revised the manuscript and received suggestions and input from the reviewer. The manuscript is ready for acceptance.